# Influence of the Nocturnal Effect on the Estimated Global CO$_2$ Flux

Rui Jin [1], Tan Yu [1,*], Bangyi Tao [2], Weizeng Shao [1,3], Song Hu [1] and Yongliang Wei [1]

[1] College of Marine Sciences, Shanghai Ocean University, Shanghai 201306, China; m200200561@st.shou.edu.cn (R.J.); shaoweizeng@mail.tsinghua.edu.cn (W.S.); shu@shou.edu.cn (S.H.); yl-wei@shou.edu.cn (Y.W.)

[2] State Key Laboratory of Satellite Ocean Environment Dynamics, Second Institute of Oceanography, Ministry of Natural Resources, Hangzhou 310012, China; taobangyi@sio.org.cn

[3] National Satellite Ocean Application Service, Ministry of Natural Resources, Beijing 100081, China

\* Correspondence: tyu@shou.edu.cn; Tel.: +86-021-6190-0169

**Abstract:** We found that significant errors occurred when diurnal data instead of diurnal–nocturnal data were used to calculate the daily sea-air CO$_2$ flux ($F$). As the errors were mainly associated with the partial pressure of CO$_2$ in seawater ($pCO_{2w}$) and the sea surface temperature ($SST$) in the control experiment, $pCO_{2w}$ and $SST$ equations were established, which are called the nocturnal effect of the CO$_2$ flux. The root-mean-square error between the real daily CO$_2$ flux ($F_{real}$) and the daily CO$_2$ flux corrected for the nocturnal effect ($F_{com}$) was 11.93 mmol m$^{-2}$ d$^{-1}$, which was significantly lower than that between the $F_{real}$ value and the diurnal CO$_2$ flux ($F_{day}$) (46.32 mmol m$^{-2}$ d$^{-1}$). Thus, the errors associated with using diurnal data to calculate the CO$_2$ flux can be reduced by accounting for the nocturnal effect. The mean global daily CO$_2$ flux estimated based on the nocturnal effect and the sub-regional $pCO_{2w}$ algorithm ($cor\_F_{com}$) was −6.86 mol m$^{-2}$ y$^{-1}$ (September 2020–August 2021), which was greater by 0.75 mol m$^{-2}$ y$^{-1}$ than that based solely on the sub-regional $pCO_{2w}$ algorithm ($day\_F_{com}$ = −7.61 mol m$^{-2}$ y$^{-1}$). That is, compared with $cor\_F_{com}$, the global $day\_F_{com}$ value overestimated the CO$_2$ sink of the global ocean by 10.89%.

**Keywords:** daytime data; CO$_2$ flux; partial pressure; nocturnal effect; ocean sink





## 1. Introduction

Since the beginning of the Industrial Revolution, human activities such as fossil fuel combustion, cement production, and land-use change have released large amounts of carbon dioxide (CO$_2$) into the atmosphere, thus disrupting the global carbon cycle and causing global climate change [1]. As an important reservoir of carbon, the oceans currently absorb approximately 25% of anthropogenic CO$_2$ emissions [2]. Although this could reach 70–80% on a timescale of a few hundred years and 80–95% on a timescale of a few thousand years, these estimates remain uncertain [3]. Some studies have suggested that the estimated errors associated with the partial pressure of CO$_2$ ($pCO_2$) are mainly at the regional level, corresponding to a difference of >10% of the mean climatic $pCO_2$, which is an order of magnitude greater than the uncertainty associated with the most advanced measurements. Yu (2014) found that a different CO$_2$ transfer velocity led to considerable uncertainty in the estimated global CO$_2$ flux [4]. Therefore, it is critical to reduce the uncertainty associated with the estimated oceanic CO$_2$ flux to improve our understanding of the potential processes that control the distribution of anthropogenic CO$_2$ between the atmosphere, land, and oceans in the present and future [5].

At present, the sea–air CO$_2$ flux can be measured directly using the eddy correlation method. Alternatively, the CO$_2$ flux is often calculated by the block method formula [4], as follows: sea–air CO$_2$ flux = sea–air gas transfer velocity × solubility of CO$_2$ in seawater × ($pCO_2$ in seawater–$pCO_2$ in air). If the CO$_2$ flux is positive, it means that CO$_2$

enters the atmosphere from the ocean, i.e., the ocean is the source of $CO_2$. If the$CO_2$ flux is negative, it means that $CO_2$ enters the ocean from the atmosphere, i.e., the ocean is the sink of $CO_2$. These parameters are obtained by remote sensing.

The algorithm for determining the $pCO_2$ of seawater based on remote sensing data mainly depends on the sea temperature (*SST*) and chlorophyll-a (*Chl-a*) concentration. Bai et al. (2015) used the relationship between these factors and the $pCO_2$ of seawater to establish the corresponding algorithm [6]. As *SST* and *Chl-a* data are mainly obtained using optical remote-sensing techniques, there are no nocturnal data; however, some researchers consider that the diurnal–nocturnal variations in *SST* and *Chl-a* are significant.

Stuart-Menteth et al. (2003) and Genemann et al. (2003) analysed *SST* data measured at mooring buoys and observed a significant daily variation in *SST*, which may have been due to the diurnal–nocturnal variation in solar radiation, wind stress, and cloud cover [7–9]. Lu (2007) observed a positive correlation between the daily variations in the $pCO_2$ of seawater and the *SST* [10]. Jeffery et al. (2007) found that the daily variation in the *SST* significantly affected the sea–air exchange of $CO_2$, increasing the emission of air from the ocean and reducing the $pCO_2$ of seawater, especially at the equator. The *SST* affects the $CO_2$ flux by influencing the $pCO_2$ of seawater and the solubility of $CO_2$ at low wind speeds [9,11]. When the reference temperature is 20 °C, the effect of the *SST* on solubility accounts for ~2.7% of the total variation in the $CO_2$ flux [12]. At high latitudes, as the solubility of $CO_2$ increases at low temperatures, the daily variation in salinity alters the ability of the oceans to absorb atmospheric $CO_2$ [13].

Marrec et al. (2014) and Borges et al. (1999) concluded that the tidal cycle affected the daily variation in phytoplankton abundance, and thus the daily variation in the $pCO_2$ of seawater [14,15]. Bates et al. (2001) argued that the extremely high productivity of organisms in coral reef ecosystems could also cause large daily variations in the $pCO_2$ of seawater [16]. Moreover, the daily variation in the $pCO_2$ of seawater is influenced by biological activity, whereby $CO_2$ is mainly consumed as a result of photosynthesis during the day and released due to respiration at night [17]. Marrec et al. (2014) estimated that the mean diurnal–nocturnal variation in the $pCO_2$ associated with the biological cycle accounted for 16% of the mean $CO_2$ sink [14].

In addition to *SST* and biological activity, Kuss et al. (2006) found that the water mass mixing process was one of the main factors controlling the variation in the $pCO_2$ of surface seawater, while the daily variation in the wind speed affected the water mass mixing process [18–21]. Jeffery et al. (2007) found that the diurnal–nocturnal variation in seawater convection also affected the sea–air $CO_2$ transfer velocity and the daily variation in the sea–air $CO_2$ flux [11,22,23]. Rousseau et al. (2020) observed that the daily variation in the atmospheric $CO_2$ concentration directly affected the $pCO_2$ of seawater [24]. Furthermore, the change in the $pCO_2$ of air affected the $CO_2$ flux. Figure 1 depicts the effects of these factors on the sea–air $CO_2$ flux.

As there is a clear diurnal–nocturnal variation in the $pCO_2$ of seawater, it is inaccurate to use solely diurnal data instead of diurnal–nocturnal data. One of the goals of this study was that the relationship between the diurnal $pCO_2$ and nocturnal $pCO_2$ was determined and used to revise the $pCO_2$ calculated based on diurnal data only. In addition to this, it is also our goal to determine the relationships between diurnal and nocturnal data for the other parameters involved in the $CO_2$ flux block method and to use the corresponding relationships to correct the diurnal data for each parameter. Ultimately improving the accuracy of the global $CO_2$ flux estimates by considering the diurnal variation of parameters.

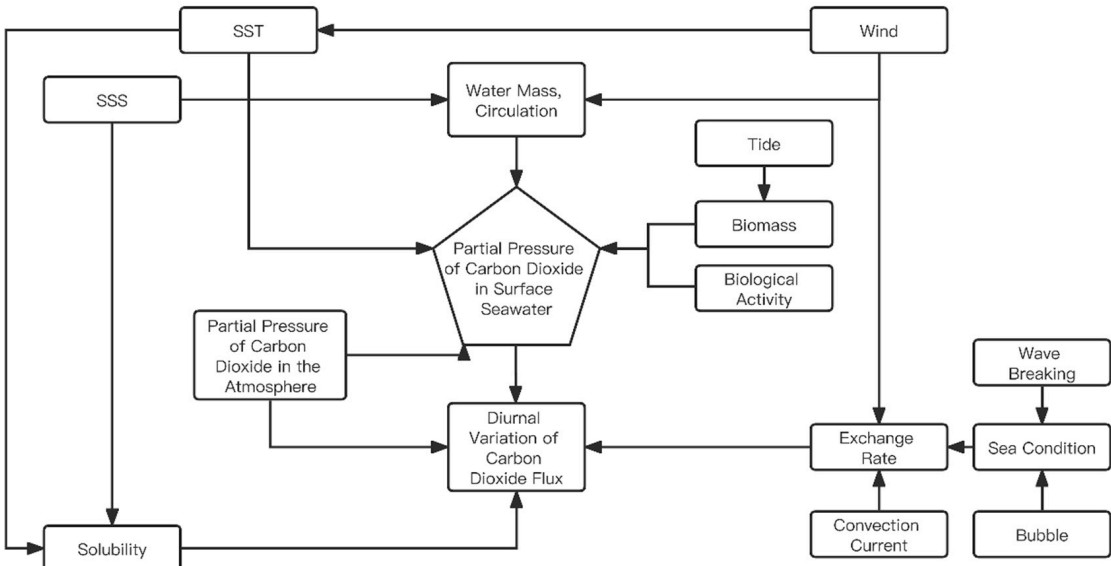

**Figure 1.** Schematic of the factors influencing the sea–air $CO_2$ flux.

## 2. Data and Methods

### 2.1. Buoy Data

The $pCO_2$, $SST$, and sea surface salinity ($SSS$) data used in this study were obtained from the global $CO_2$ time series and mooring project of the Ocean Carbon Data System (OCADS) (https://www.ncei.noaa.gov/access/ocean-carbon-data-system/oceans/time_series_moorings.html, accessed on 8 May 2022). International organisations from 18 countries have installed sensors on moored buoys to provide high-resolution time series measurements of the $pCO_2$ of the atmospheric boundary layer and ocean surface. Time series and mooring projects on $CO_2$ are coordinated by the International Ocean Carbon Coordination Project (IOCCP) and OceanSITES.

Figure 2 shows a map of the buoy stations, where data are taken at 00:00, 03:00, 06:00, 09:00, 12:00, 15:00, 18:00, and 21:00. In Figure 3, the period 2010 to 2020 has the largest number of buoy stations, so we chose this time range as the study time in our study.

### 2.2. Satellite Remote Sensing Data

#### 2.2.1. Wind Data and Atmospheric Pressure Data

Wind and atmospheric pressure data from 2010 to 2020 were obtained from ERA5 (https://cds.climate.copernicus.eu/cdsapp#!/dataset/reanalysis-era5-single-levels?tab=overview, accessed on 8 May 2022), which is the fifth-generation European Centre for Medium-Range Weather Forecasts (ECMWF) reanalysis of global climate and weather over the past 4–7 years. We used the $u$ and $v$ components of the wind speed (m s$^{-1}$) at a height of 10 m above the Earth's surface, with a time resolution of 1 h and a spatial resolution of $0.25° \times 0.25°$. To correspond to the $pCO_2$, $SST$, and $SSS$ data of the buoys, wind and atmospheric pressure data at 00:00, 03:00, 06:00, 09:00, 12:00, 15:00, 18:00, and 21:00 were selected.

#### 2.2.2. $SST$ and $Chl$-$a$ Data

The $SST$ and $Chl$-$a$ data used in this study were obtained from the Aqua MODIS global map 11-µm daytime $SST$ and $Chl$-$a$ data (version R2019.0, https://oceandata.sci.gsfc.nasa.gov/directdataaccess/Level-3%20Mapped/Aqua-MODIS, accessed on 8 May 2022) for the period June 2020 to May 2021 at a temporal resolution of 1 day and a spatial resolution of 4 km $\times$ 4 km.

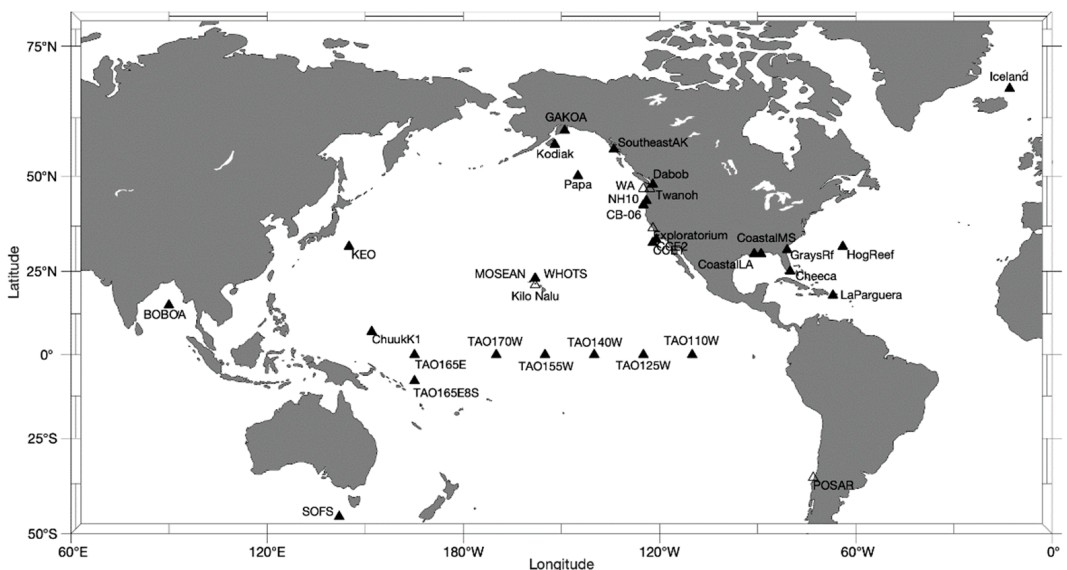

**Figure 2.** Map of global buoy stations. ▲ indicates the selected stations for this study.

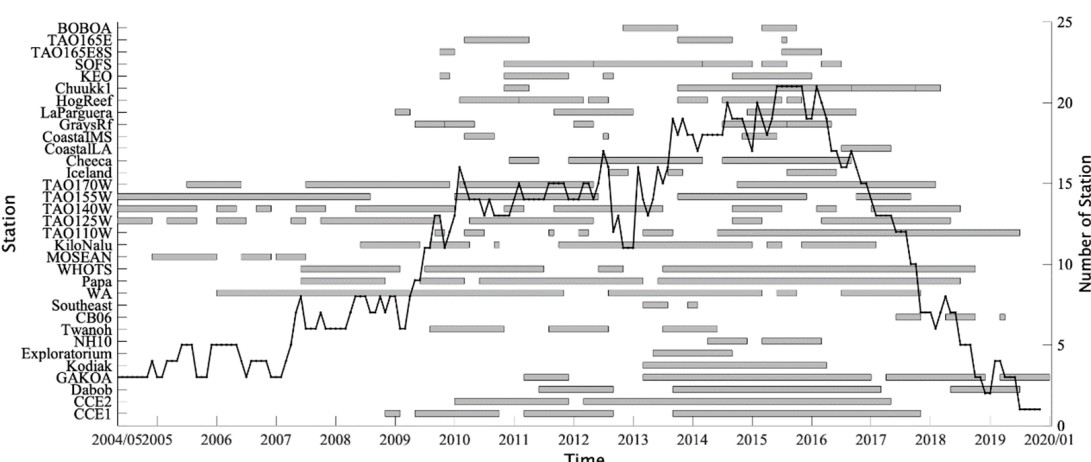

**Figure 3.** Time series of global buoy data. The right axis shows the number of stations corresponding to a given time. The horizontal bars represent the data available at the corresponding site at different times.

### 2.2.3. *SSS* Data

The *SSS* data were obtained from the 10-day 3D global ocean forecast data (spatial resolution of $0.083° \times 0.083°$), which are updated daily at 00:00, 06:00, 12:00, and 18:00 by the global ocean analysis and prediction system (https://resources.marine.copernicus.eu/product-detail/GLOBAL_ANALYSIS_FORECAST_PHY_001_024/INFORMATION, accessed on 8 May 2022).

### 2.2.4. Carbon Dioxide and Water Vapour Data

The atmospheric $CO_2$ concentration and water vapour data were obtained from Aqua AIRS IR-only Level 3 climcaps (gridded daily V2 with integrated quality control), with two daily tracks divided into diurnal and nocturnal data with a spatial resolution of $1° \times 1°$ (https://disc.gsfc.nasa.gov/datasets/SNDRAQIL3CDCCP_2/summary?keywords=CO2, accessed on 8 May 2022).

*2.3. Calculation of the $CO_2$ Flux*

The block formula of the sea–air $CO_2$ flux [25], $F$ (mmol m$^{-2}$ d$^{-1}$ or mol m$^{-2}$ s$^{-1}$), is as follows:

$$F = kL\Delta pCO_2 \tag{1}$$

When the atmospheric $CO_2$ concentration is high, $CO_2$ moves from the atmosphere to the ocean; thus, $F$ is negative. The direction of $F$ is determined by the difference between the $pCO_2$ of seawater and air (i.e., $\Delta pCO_2$) [26], which is usually expressed in units of µatm and is calculated using Equation (2):

$$\Delta pCO_2 = pCO_{2w} - pCO_{2a} \tag{2}$$

where $pCO_{2w}$ is the $pCO_2$ of seawater (in Pa or µatm) and $pCO_{2a}$ is the $pCO_2$ of air (in Pa or µatm).

The sea–air gas transfer velocity, $k$ (cm h$^{-1}$), is expressed as follows [27]:

$$k = 0.251U_{10}^2(Sc/660)^{-0.5} \tag{3}$$

where $U_{10}$ is the wind speed (m s$^{-1}$) at a height of 10 m above sea level and $Sc = A + Bt + Ct^2 + Dt^3 + Et^4$ ($t$ is the temperature in °C; $A = 1923.6$, $B = -125.06$, $C = 4.3773$, $D = -0.085681$, and $E = 0.00070284$).

The solubility of $CO_2$ in seawater, $L$ (mol L$^{-1}$ atm$^{-1}$), was calculated using Weiss' formula [28]:

$$\ln L = A_1 + A_2(100/SST) + A_3 \ln(SST/100) + \\ SSS‰[B_1 + B_2(SST/100) + B_3(SST/100)^2] \tag{4}$$

where $SST$ is the absolute $SST$ (in K) (absolute $SST = t$ (°C) + 273.15), $SSS$ is the surface seawater salinity, $A_1 = -58.0931$, $A_2 = 90.5069$, $A_3 = 22.294$, $B_1 = 0.027766$, $B_2 = -0.025888$, and $B_3 = 0.0050578$.

## 3. Results and Discussion

### 3.1. Estimated Daily Variation in the $CO_2$ Flux

Figure 4 shows that there was a significant diurnal–nocturnal variation in the sea–air $CO_2$ flux. As the sea–air $CO_2$ flux is usually estimated using diurnal remote sensing data, we studied the difference between the $CO_2$ flux calculated using (i) diurnal data ($F_{day}$) only and (ii) diurnal–nocturnal data ($F_{real}$). There was a significant difference between the $F_{day}$ and $F_{real}$ values (Figure 5). The largest difference was observed at HogReef station (64°W, 32°N), where $F_{real}$ was 4.31 mmol m$^{-2}$ d$^{-1}$ lower than $F_{day}$ on average. In contrast, the smallest difference was observed at BOBOA station (90°E, 15°N), where $F_{real}$ was 0.01 mmol m$^{-2}$ d$^{-1}$ lower than $F_{day}$. Of the stations where $F_{real}$ was larger than $F_{day}$, CoastalMS (88°W, 30°N) had the largest $F_{real} - F_{day}$ value of 2.64 mmol m$^{-2}$ d$^{-1}$. Temporally, the largest difference was observed in 2018 (data for 2020 were sparse and not included in the comparison), whereas the smallest difference was observed in 2011. The largest difference was observed on 27 August 2018, when $F_{real}$ was 21.90 mmol m$^{-2}$ d$^{-1}$ lower than $F_{day}$. The smallest difference was observed on 27 July 2011, when $F_{real}$ was $1.69 \times 10^{-5}$ mmol m$^{-2}$ d$^{-1}$ higher than $F_{day}$. The average difference across the period from 2010 to 2020 was 0.16 mmol m$^{-2}$ d$^{-1}$. Therefore, using diurnal data instead of diurnal–nocturnal data to calculate the $CO_2$ flux will cause significant errors in the calculation of the daily $CO_2$ flux. Accordingly, this study attempts to eliminate such errors.

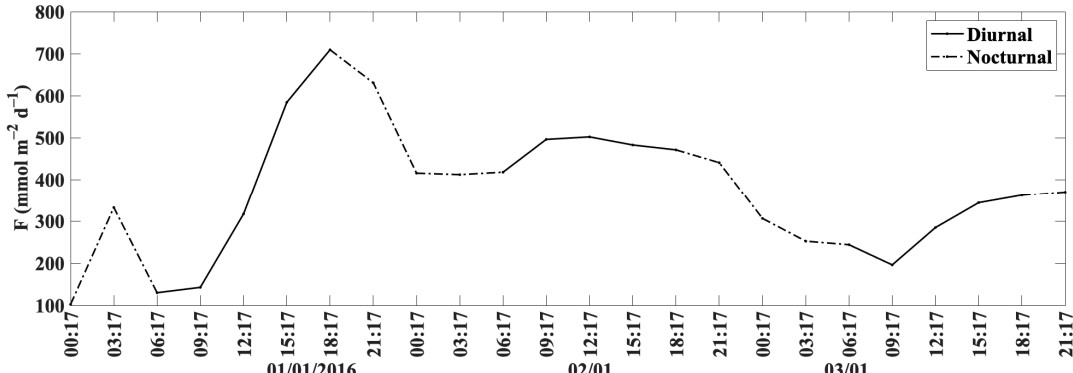

**Figure 4.** Sea–air $CO_2$ flux at station Kodiak (152°W, 57°N) from 1–3 January 2016. Diurnal hours are 06:00, 09:00, 12:00, and 15:00, and nocturnal hours are 18:00, 21:00, 00:00, and 03:00.

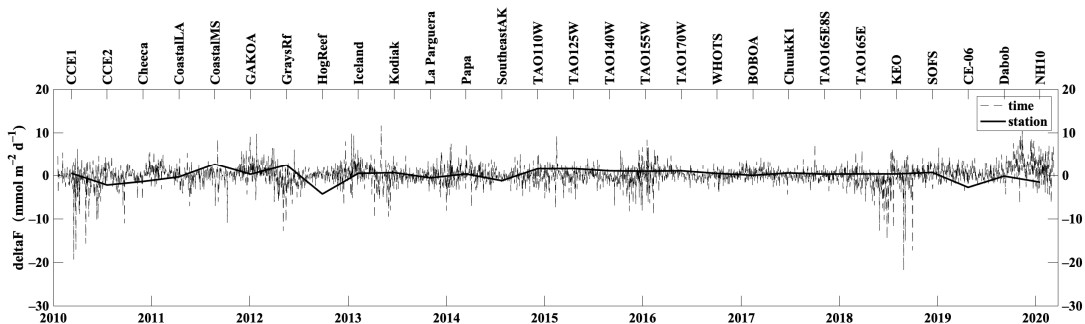

**Figure 5.** Average difference in the $CO_2$ flux calculated with and without the nocturnal effect at global buoy stations, with $deltaF = F_{real} - F_{day}$, the upper abscissa as the names of the global stations, and the lower abscissa as time. – shows the value corresponding to the time of the horizontal coordinate; — shows the value corresponding to the station of the horizontal coordinate.

### 3.2. Control Experiment on the Daily $CO_2$ Flux

To understand the main factors controlling the difference between $CO_2$ fluxes calculated using diurnal data and those calculated using diurnal–nocturnal data, a single-factor control experiment was conducted using buoy data from 2010 to 2020.

In the control experiment, the diurnal *SST*, *SSS*, wind speed, $pCO_{2w}$, and $pCO_{2a}$ data were used to calculate the daily $CO_2$ flux, thus obtaining $F_{SST}$, $F_{SSS}$, $F_{k_{660}}$, $F_{pCO_{2w}}$, and $F_{pCO_{2a}}$, respectively, where $k_{660}$ is the gas transfer velocity $k$ calculated using *Sc* of seawater at 20 °C (*Sc* = 660) and wind speed data. In each single-factor control experiment, the diurnal–nocturnal data were used to calculate the daily $CO_2$ flux, but the selected influencing factor was excluded from the calculation. The results of the control experiment are shown in Figure 6. The maximum $F_{pCO_{2w}} - F_{real}$ value from 2010 to 2020 was 1.21 mmol m$^{-2}$ d$^{-1}$. The $F_{k_{660}} - F_{real}$ value, which indicated the influence of the daily variation in the second power of the wind speed on the calculation of the $CO_2$ flux, was also large, with a mean value of 0.312 mmol m$^{-2}$ d$^{-1}$. Using only the diurnal data of $pCO_{2a}$ to calculate the daily $CO_2$ flux also caused a considerable error of 0.157 mmol m$^{-2}$ d$^{-1}$. The daily variation in *SSS* strongly affected the daily variation in *L*; however, this had little effect on the daily variation in the $CO_2$ flux. The influence of *SST* on *L* and *Sc* did not have a significant effect on the daily variation in the $CO_2$ flux (Figure 6). However, *SST* strongly influenced the daily variation in $pCO_{2w}$, and in turn $pCO_{2w}$ strongly influenced the daily variation in the $CO_2$ flux; therefore, *SST* significantly affected the diurnal variation in the $CO_2$ flux.

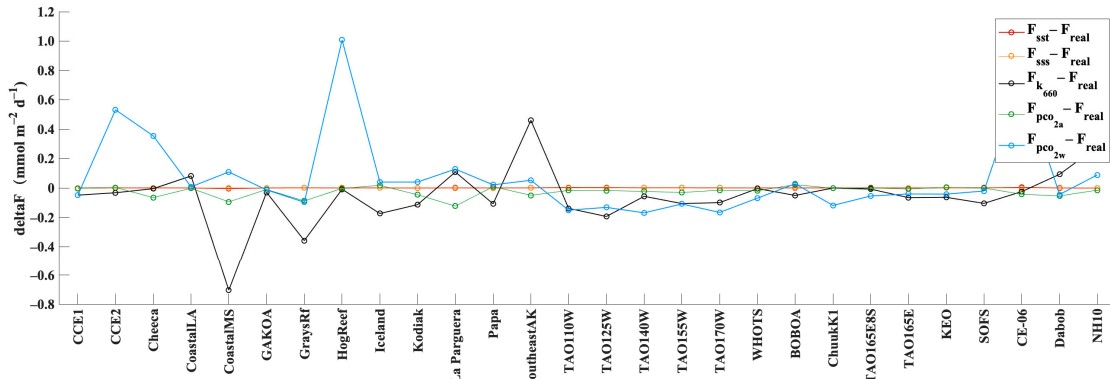

**Figure 6.** Effects of single factors on the calculated $CO_2$ flux at global stations from 2010 to 2020. The vertical coordinate is the difference between Freal and the $CO_2$ flux calculated after controlling for a single influencing factor.

As shown in Figure 6, there were clear differences between the $F_{pCO_{2w}}$ and $F_{real}$ values at stations CCE2 (121°W, 34°N), Cheeca (80°W, 25°N), HogReef (64°W, 32°N), and CE-06 (125°W, 43°N). These stations were selected to consider the influence of each single factor on the calculation of the daily $CO_2$ flux over time. As shown in Figure 7, data from HogReef station covered the period from August 2016 to July 2018. The maximum and minimum $F_{pCO_{2w}} - F_{real}$ values were 21.77 mmol m$^{-2}$ d$^{-1}$ and $1.66 \times 10^{-2}$ mmol m$^{-2}$ d$^{-1}$, respectively. The daily $CO_2$ flux that was calculated using the diurnal $pCO_{2w}$ data only corresponded to an overall decrease (increase) in the $CO_2$ sink (source) of the ocean; thus, the correction of $pCO_{2w}$ resulted in a larger oceanic $CO_2$ sink and smaller oceanic $CO_2$ source values. The $F_{k_{660}} - F_{real}$ value exhibited an obvious seasonal variation, being smaller during October–November and May–July, with a minimum value of $-7.75 \times 10^{-4}$ mmol m$^{-2}$ d$^{-1}$. Relatively large $CO_2$ fluxes were observed from December to April and from August to September, with a maximum of $-26.71$ mmol m$^{-2}$ d$^{-1}$. Only diurnal wind data were used to calculate the daily $CO_2$ flux, which corresponded to increases in the $CO_2$ source and sink of the ocean. The sink value increased more than the source value.

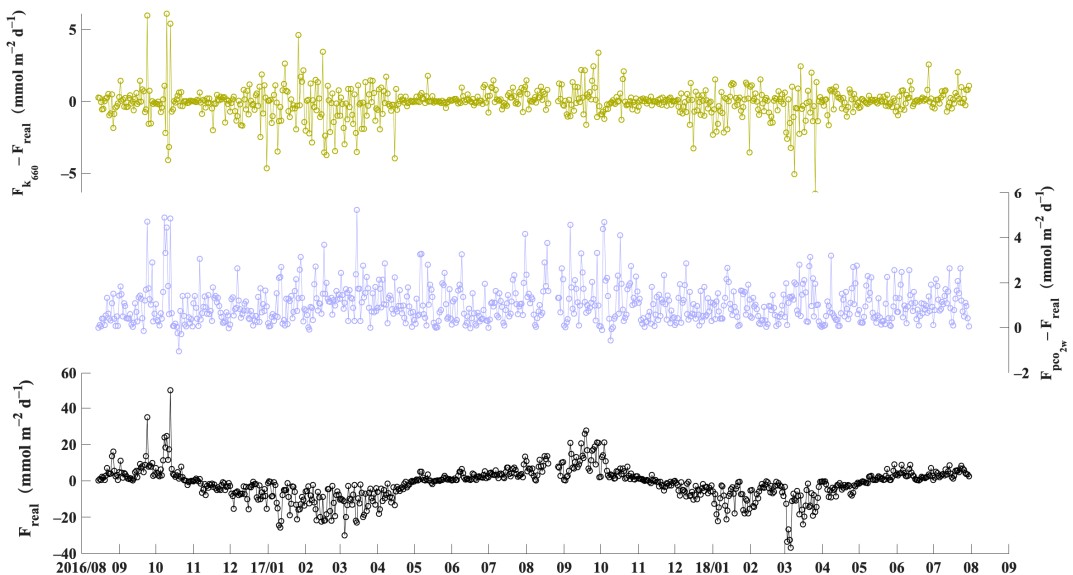

**Figure 7.** Differences in experimental $CO_2$ fluxes ($F_{k_{660}} - F_{real}$ and $F_{pCO_{2w}} - F_{real}$) and $F_{real}$ at HogReef station from August 2016 to July 2018.

There were also significant differences between the $F_{k_{660}}$ and $F_{real}$ values at stations CoastalMS (88°W, 30°N), GraysRf (81°W, 31°N), SoutheastAK (134°W, 56°N), and NH10 (124°W, 44°N). The results of the control experiment at SoutheastAK, where the difference

between $F_{k_{660}}$ and $F_{real}$ was large and the time series had the longest continuity, revealed that the influence of each factor on the error in the daily $CO_2$ flux calculation exhibited obvious seasonal differences. The $F_{k_{660}} - F_{real}$ values were lower from September to October and in March, with a minimum of $-1.90 \times 10^{-3}$ mmol m$^{-2}$ d$^{-1}$, whereas higher values were observed from April to August and from November to February, with a maximum of 97.70 mmol m$^{-2}$ d$^{-1}$. Although the $CO_2$ flux calculated using the diurnal data of each influencing factor was either larger or smaller than the daily $CO_2$ flux, with an obvious seasonal variation, this difference was not observable at all stations. When the diurnal data of each influencing factor were used to calculate the $CO_2$ flux, the calculated daily $CO_2$ flux from June to September increased at some stations, whereas it decreased at other stations. This was also the case from October to December and from January to May.

The daily variation in $pCO_{2w}$ had a considerable influence on the daily variation in the $CO_2$ flux, and the $SST$ value strongly influenced the daily variation in the $CO_2$ flux by affecting $pCO_{2w}$ (when $SSS$ was not considered). Although the daily variation in the wind speed also had a significant effect on the daily variation in the $CO_2$ flux, wind speed was not considered when establishing the nocturnal effect relationship because 24 h wind data were generally available. Therefore, it is recommended to use diurnal–nocturnal wind data to calculate the daily mean wind speed, and not to use the daytime wind data instead.

### 3.3. Nocturnal Effect Relationship

To eliminate the error caused by using diurnal data instead of diurnal–nocturnal data to calculate the $CO_2$ flux, we studied the relationship between diurnal and nocturnal $CO_2$ fluxes. The relationship between diurnal and nocturnal $pCO_{2w}$ values is termed the nocturnal effect of $pCO_{2w}$, and the relationship between diurnal and nocturnal $SST$ value is termed the nocturnal effect of $SST$. The nocturnal effects of $pCO_{2w}$ and $SST$ are collectively termed the nocturnal effect of the $CO_2$ flux. Diurnal and nocturnal $CO_2$ fluxes were calculated using diurnal and nocturnal data from various stations worldwide. The correlation coefficients between the calculated diurnal and nocturnal $CO_2$ fluxes were determined using a 99.9% significance test. As shown in Figure 8, the diurnal and nocturnal $CO_2$ fluxes were significantly correlated, with a correlation coefficient of 0.998 at station TAO155W (155°W, 0°N) in the Pacific Ocean. The weakest correlation (0.953) was observed at station NH10 (124°W, 44°N) in the Pacific Ocean. No obvious regional characteristics were observed between the location of stations in the global ocean (Figure 8) and the correlation coefficients between their diurnal–nocturnal mean $CO_2$ fluxes. Moreover, the correlation coefficients differed between proximate stations.

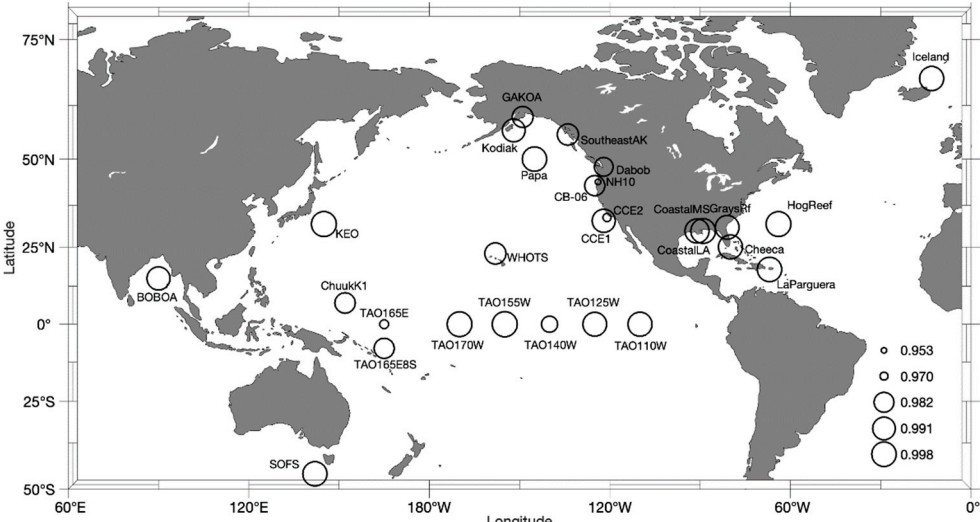

**Figure 8.** Spatial distribution of correlation coefficients between calculated diurnal and nocturnal $CO_2$ fluxes.

- Nocturnal effect of the $pCO_2$ of seawater

The nocturnal effect on the $pCO_{2w}$ value was obtained from the fitting results in Figure 9a:

$$pCO_{2wn} = Y_1 \times pCO_{2wd} + Y_2 \tag{5}$$

where $pCO_{2wn}$ is the nocturnal $pCO_2$ of seawater (μatm), $pCO_{2wd}$ is the diurnal $pCO_2$ of seawater (μatm), $Y_1 = 0.9898$, and $Y_2 = 3.0999$.

- Nocturnal effect of *SST*

The nocturnal effect on the *SST* value was obtained from the fitting results in Figure 9b:

$$SST_n = Z_1 \times SST_d + Z_2 \tag{6}$$

where $SST_n$ is the nocturnal *SST* (°C), $SST_d$ is the diurnal *SST* (°C), $Z_1 = 1.0012$, and $Z_2 = 0.0753$.

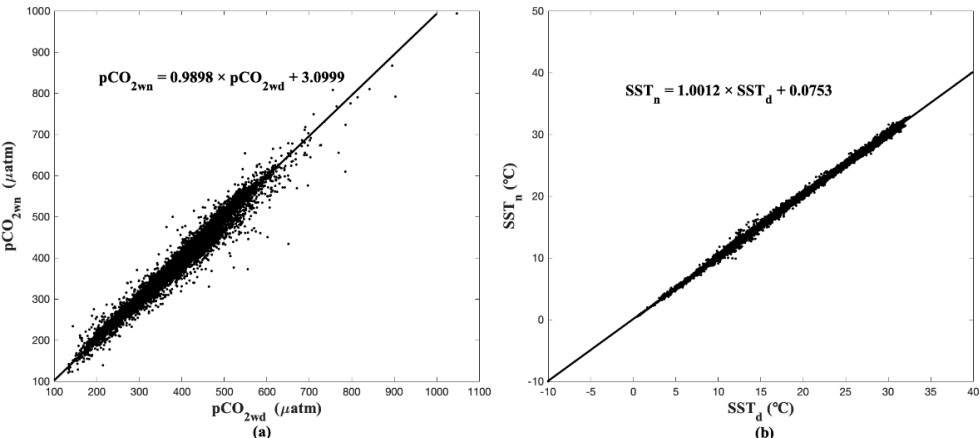

**Figure 9.** Fitting results of (**a**) nocturnal $pCO_{2w}$ ($pCO_{2wn}$) and diurnal $pCO_{2w}$ ($pCO_{2wd}$), and (**b**) nocturnal *SST* ($SST_n$) and diurnal *SST* ($SST_d$), whereby fitting results of using 75% of the data from 2010 to 2020.

- Daily variation in *Chl-a*

The *Chl-a* data from the Kiyomoto Yoko experiment (2003) are scarce and have little temporal continuity, and we chose the data with the longest temporal continuity to plot Figure 10. As no diurnal–nocturnal rule in *Chl-a* was observed (Figure 10), the nocturnal effect of *Chl-a* was not considered in this study. The *Chl-a* data is limited, so the conclusions may not be representative, and more *Chl-a* diurnal-nocturnal data is needed to support this conclusion. We couldn't obtain the nocturnal effect formula of *Chl-a* similar to *SST* (Equation (6)). So, we directly considered the nocturnal effects of $pCO_{2w}$. There were two obvious changes in the curve, which probably related to the change in the sampling station during the *Chl-a* experiment.

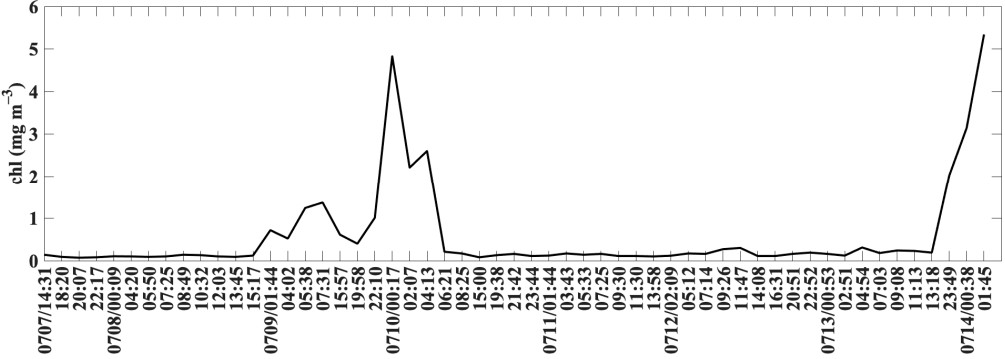

**Figure 10.** *Chl-a* data from the Kiyomoto_Yoko experiment (7–14 July 2003).

### 3.4. Comparison of Calculated and Real Daily $CO_2$ Fluxes

Equation (5) and $pCO_{2wd}$ were used to calculate $pCO_{2wn}$, and the diurnal–nocturnal data of $SSS$, wind speed, $pCO_{2a}$, and $SST_d$ were used to calculate the diurnal–nocturnal $CO_2$ flux ($F_{comp}$). In addition, Equation (6) and $SST_d$ were used to calculate $SST_n$, and the diurnal–nocturnal data of $SSS$, wind speed, $pCO_{2a}$, and $pCO_{2wd}$ were used to calculate the diurnal–nocturnal $CO_2$ flux ($F_{comt}$). By using Equations (5) and (6), $SST_n$ and $pCO_{2wn}$ were calculated based on $SST_d$ and $pCO_{2wd}$, respectively, and the daily $CO_2$ flux was calculated by combining the diurnal–nocturnal data of $SSS$, wind speed, and $pCO_{2a}$ ($F_{com}$). The $F_{com}$, $F_{comp}$, and $F_{comt}$ values were compared with the $F_{real}$ data using the root-mean-square error ($RMSE$):

$$RMSE = \sqrt{\frac{\sum_{i=1}^{n}[comF - F_{real}]^2}{n}} \qquad (7)$$

where *comF* is $F_{comt}$, $F_{comp}$, or $F_{com}$; $F_{real}$ is the real daily $CO_2$ flux; and *n* is the number of data observations.

The results are shown in Figure 11, where $F_{comp}$ is overlapped by $F_{com}$ because the difference between $F_{comp}$ and $F_{com}$ was very small. The *RMSE* values between $F_{real}$ and $F_{comt}$, $F_{comp}$, $F_{com}$, and $F_{day}$ were 12.58 mmol m$^{-2}$ d$^{-1}$, 11.94 mmol m$^{-2}$ d$^{-1}$, 11.93 mmol m$^{-2}$ d$^{-1}$, and 46.32 mmol m$^{-2}$ d$^{-1}$, respectively. Thus, compared with $F_{day}$, the values of $F_{comt}$, $F_{comp}$, and $F_{com}$ were more accurate and closer to $F_{real}$. The similar *RMSE* of $F_{comt}$, $F_{comp}$, and $F_{com}$ indicate that there was a coincidence between the nocturnal effects of $pCO_{2w}$ and $SST$. As $SST$ is the most important influencing factor of $pCO_{2w}$, it is an important parameter for establishing the algorithm of $pCO_{2w}$.

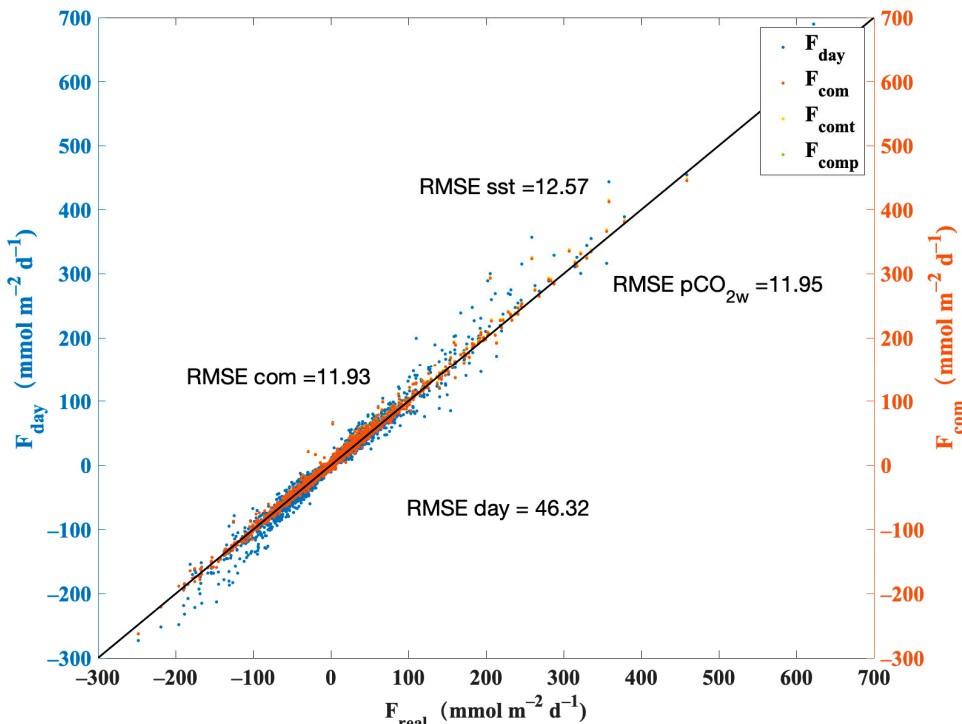

**Figure 11.** Results of using 25% of the data from 2010 to 2020 verify the calculated nocturnal effect.

### 3.5. Estimated Global $CO_2$ Flux

#### 3.5.1. $pCO_2$ Remote Sensing Inversion Algorithm

As the remote sensing data of the $SST$ and *Chl-a* parameters that correspond to the algorithm are solely diurnal, $pCO_{2wd}$ and $SST_d$ were used to develop a global $pCO_{2w}$ algorithm as follows:

$$pCO_{2wd} = W_1 \times SST_d + W_2 \times ln(Chl\text{-}a) + W_3 \qquad (8)$$

where $SST_d$ is the absolute daily $SST$ (°C) and $Chl$-$a$ is the $Chl$-$a$ concentration (mg m$^{-3}$) at the sea surface.

According to the fitting results in Figure 12a, $W_1 = 3.40$ in the $pCO_{2wd}$ calculation model. The influence of $SST$ on $pCO_{2wd}$ was removed to obtain n$pCO_{2wd}$. According to the fitting results in Figure 12b, $W_2 = -4.44$ and $W_3 = 325.11$ in the $pCO_{2wd}$ calculation model.

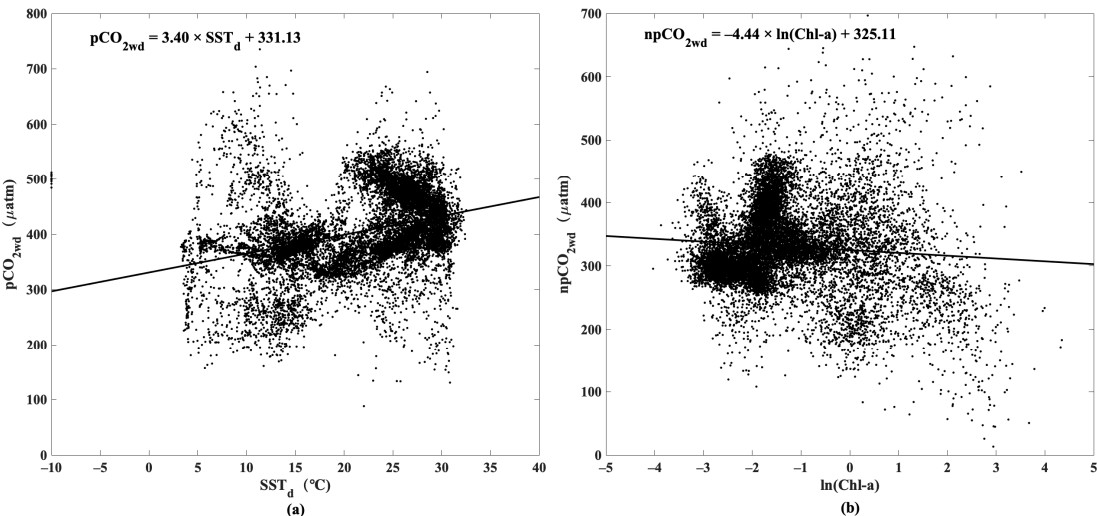

**Figure 12.** Fitting results of the global algorithm (2010–2020) between (**a**) $pCO_{2wd}$ and $SST_d$, and (**b**) $pCO_{2wd}$ with the temperature effect removed (n$pCO_{2wd}$) and $Chl$-$a$.

Using all the buoy data, a $pCO_{2wd}$ calculation model was established. The correlation coefficient between $pCO_{2wd}$ and $SST_d$ was 0.327 and passed the 99.9% significance test. The correlation coefficient between n$pCO_{2wd}$ and $Chl$-$a$ was 0.238 and also passed the 99.9% significance test. As the fitting effect was poor, the Pacific Ocean, Atlantic Ocean, and Indian Ocean sub-regions were selected to establish the calculation model.

According to the results in Figure 13, $W_1 = 3.67$, $W_2 = 8.58$, and $W_3 = 346.94$ in the $pCO_{2wd}$ model of the Pacific Ocean sub-region. The correlation coefficient between $pCO_{2wd}$ and $SST_d$ was 0.369, while that between n$pCO_{2wd}$ and $Chl$-$a$ was $-0.143$. Both passed the 99.9% significance test. For the $pCO_{2wd}$ model of the Atlantic Ocean sub-region, $W_1 = 6.28$, $W_2 = -11.48$, and $W_3 = 231.98$. The correlation coefficient between $pCO_{2wd}$ and $SST_d$ was 0.413, whereas that between n$pCO_{2wd}$ and $Chl$-$a$ was $-0.392$. Both passed the 99.9% significance test. For the $pCO_{2wd}$ model of the Indian Ocean sub-region, $W_1 = 12.96$, $W_2 = 0$, and $W_3 = 12.54$. The correlation coefficient between $pCO_{2wd}$ and $SST_d$ was 0.826 and passed the 99.9% significant test; however, $pCO_{2wd}$ was not correlated with $Chl$-$a$. Although the $pCO_{2wd}$ model performed well for the Indian Ocean sub-region, the Pacific and Atlantic Ocean sub-regions had the strongest influence on the global $pCO_{2wd}$ model.

### 3.5.2. Estimation of the CO$_2$ Flux Using the Nocturnal Effect

Remote sensing data of $SST_d$ and $Chl$-$a$ were used to calculate the $pCO_{2wd}(com\_pCO_{2wd})$ for the $pCO_{2wd}$ sub-region calculation model. In addition, $com\_pCO_{2wd}$ was combined with the remote sensing data of $SST_d$ and the diurnal data of $SSS$, $pCO_{2a}$ and wind speed data were used to calculate the diurnal CO$_2$ flux ($day\_F_{com}$). The corresponding ($com\_pCO_{2wn}$) was calculated using Equation (5) and $com\_pCO_{2wd}$, whereas the corresponding $SST_n$ was calculated using Equation (6) and $SST_d$. Combining $com\_pCO_{2wd}$, $com\_pCO_{2wn}$, $SST_d$, and $SST_n$ with the diurnal–nocturnal data of $SSS$, $pCO_{2a}$, and wind speed, the CO$_2$ flux considering the nocturnal effect and $pCO_{2wd}$ calculation model ($cor\_F_{com}$) was calculated. The distribution of $cor\_F_{com} - day\_F_{com}$ is shown in Figure 14. The $cor\_F_{com}$ value was smaller than the $day\_F_{com}$ value at low latitudes, whereas it was greater at high latitudes. The $cor\_F_{com} - day\_F_{com}$ value also varied considerably with latitude, being smaller and greater at low and high latitudes, respectively.

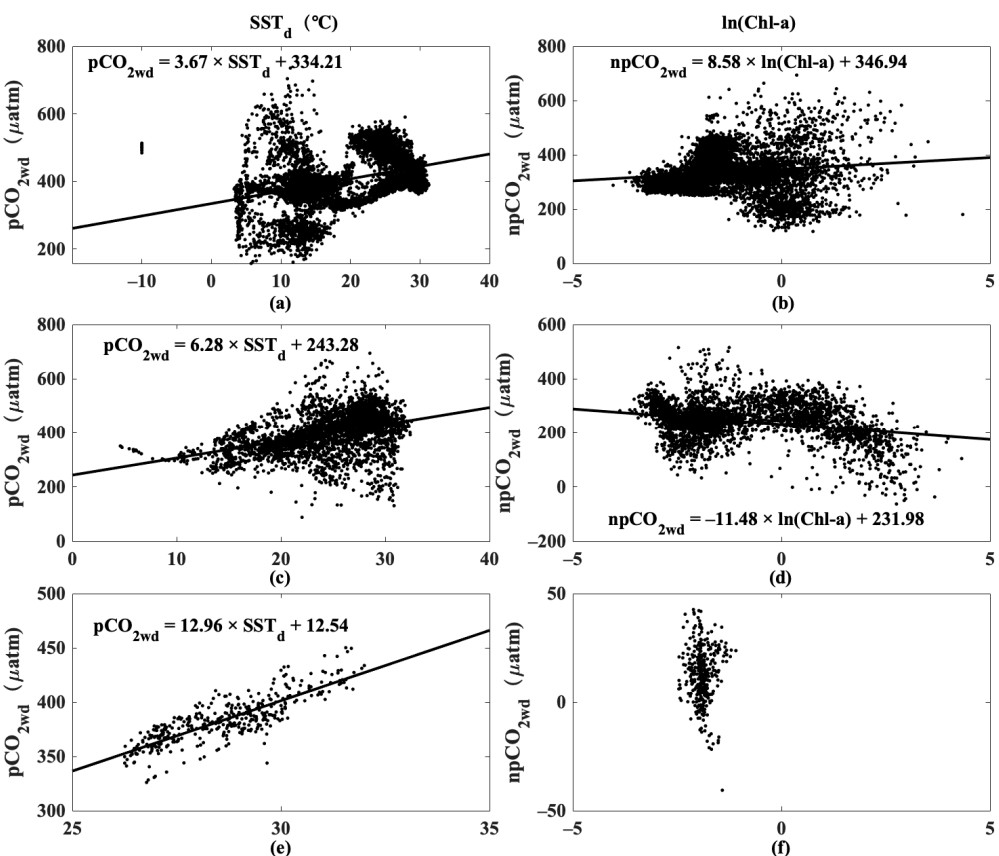

**Figure 13.** Regional algorithm for 2010–2020, showing the fitting results between $pCO_{2wd}$ and $SST_d$ in the (**a**) Pacific, (**c**) Atlantic, and (**e**) Indian Ocean sub-regions; and between n$pCO_{2wd}$ and *Chl-a* in the (**b**) Pacific, (**d**) Atlantic, and (**f**) Indian Ocean sub-regions.

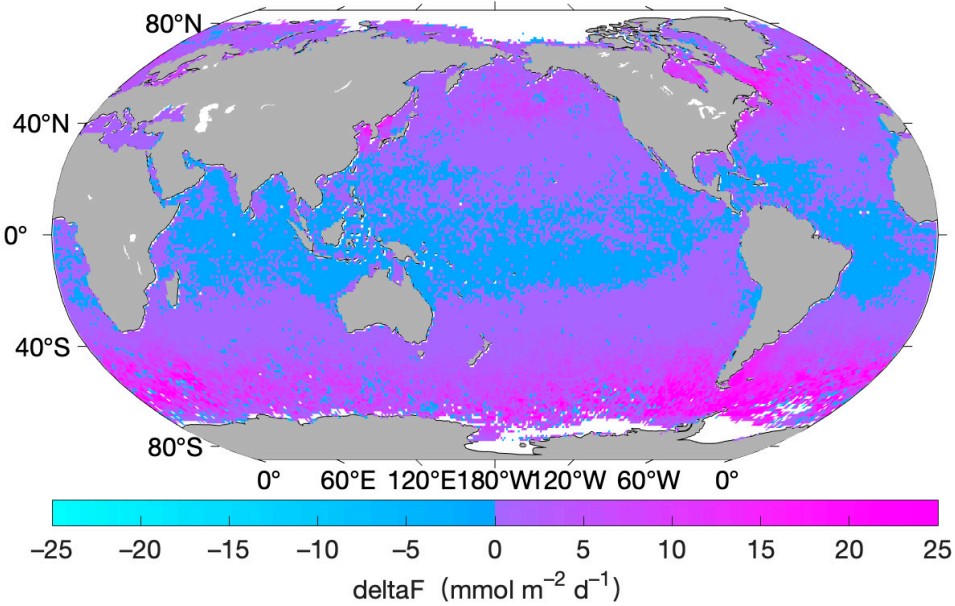

**Figure 14.** Global distribution of $cor\_F_{com} - day\_F_{com}$ (*deltaF*) from September 2020 to August 2021 (the flux calculation lacked data from 30 May to 20 June 2021 and from 22 to 27 June 2021).

As shown in Figure 14, the source and sink areas of $CO_2$ in the ocean were at low and high latitudes, respectively. The mean daily, monthly, and annual global $CO_2$ fluxes were $-4.80 \times 10^{-3}$ mmol m$^{-2}$ d$^{-1}$, $-23.36$ mmol m$^{-2}$ month$^{-1}$, and $-6.86$ mol m$^{-2}$ y$^{-1}$,

respectively, indicating that the global ocean acted as an overall sink of atmospheric $CO_2$ from September 2020 to August 2021.

As shown in Figures 14 and 15, compared with the use of $day\_F_{com}$, the use of $cor\_F_{com}$ decreased the source and sink amounts of oceanic $CO_2$. Specifically, compared with $day\_F_{com}$, the global $cor\_F_{com}$ value increased by 0.18 mmol m$^{-2}$ d$^{-1}$, thereby $day\_F_{com}$ overestimating the oceanic $CO_2$ sink by 10.21%. The mean monthly increase was 2.50 mmol m$^{-2}$ month$^{-1}$, thus $day\_F_{com}$ overestimating the mean oceanic $CO_2$ sink by 10.68%. The mean annual increase was 0.75 mol m$^{-2}$ y$^{-1}$, thereby $day\_F_{com}$ overestimating the mean oceanic $CO_2$ sink by 10.89%.

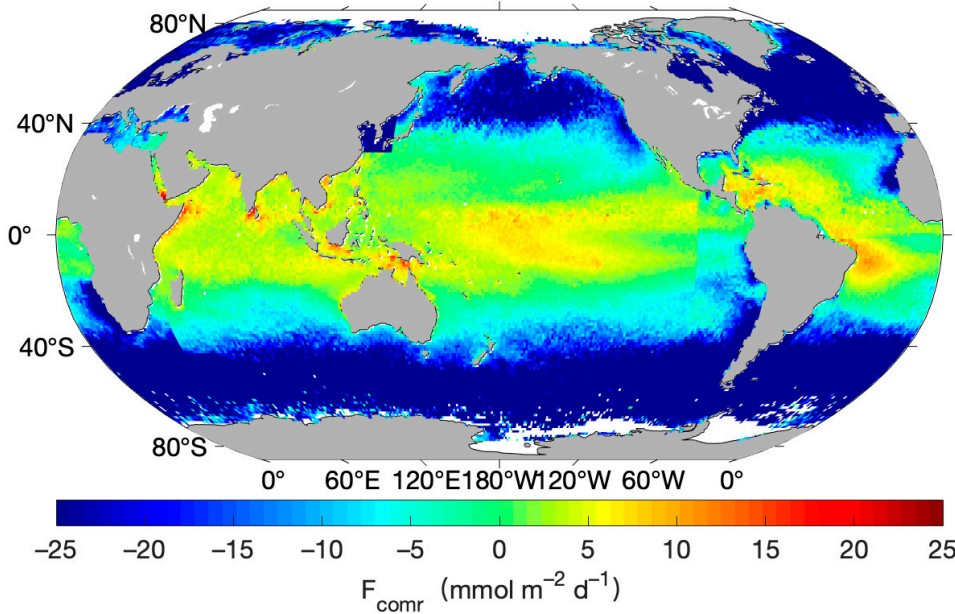

**Figure 15.** Distribution of $CO_2$ sources and sinks in the global ocean from September 2020 to August 2021.

For the convenience of understanding, we drew the flow diagram of the nocturnal effect establishment–checking–application, which is shown in Figure 16. There are many variable symbols in this paper, so we describe each of these in the accompanying Table A1.

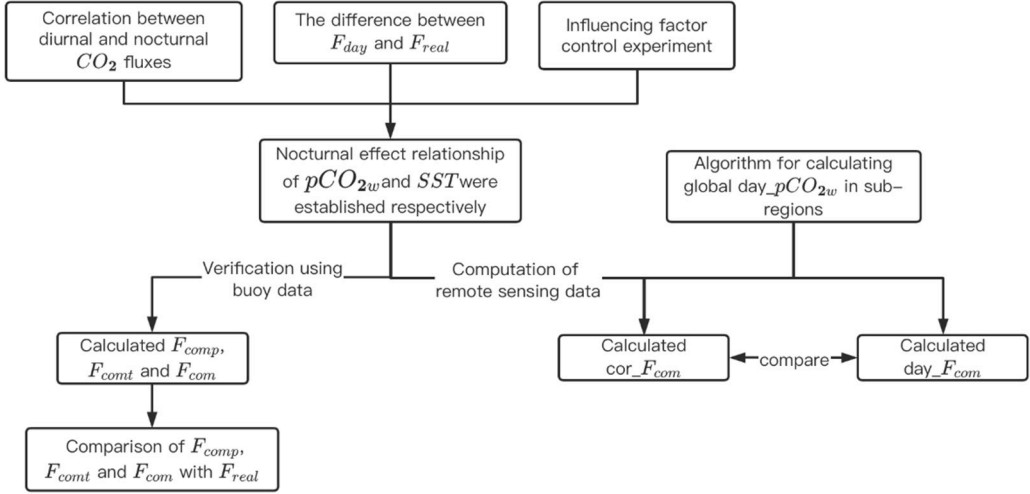

**Figure 16.** Research flow chart.

## 4. Conclusions

Calculating the daily $CO_2$ flux based on solely diurnal data of *SST*, *SSS*, wind speed, $pCO_{2w}$, and $pCO_{2a}$ instead of the corresponding diurnal–nocturnal data can lead to significant errors. In this study, the mean $F_{day} - F_{real}$ value calculated based on buoy data from 2010 to 2020 was 0.0751 mmol m$^{-2}$ d$^{-1}$. The corresponding $CO_2$ flux calculated using solely the diurnal data of *SST*, *SSS*, wind speed, $pCO_{2w}$, and $pCO_{2a}$ increased or decreased the $F_{real}$ value and exhibited obvious seasonal variations. The results of a control experiment showed that the daily variation in $pCO_{2w}$ had the greatest influence on the daily variation in the $CO_2$ flux; therefore, the *SST* value, which influences the daily variation in $pCO_{2w}$, also significantly affected the daily variation in the $CO_2$ flux.

We found that the diurnal and nocturnal $CO_2$ fluxes were significantly correlated, with correlation coefficients of >0.950 based on a 99.9% significance test. In addition, the strength of the correlation was independent of the station location. To eliminate errors associated with using diurnal data instead of diurnal–nocturnal data to calculate the $CO_2$ flux, 75% of the randomly selected buoy data from 2010 to 2020 were used and the relationship between the nocturnal effects of *SST* and $pCO_{2w}$ was established (Equations (5) and (6)). The nocturnal effect of the $CO_2$ flux was verified based on the remaining buoy data (i.e., 25%), and the *RMSE* values between $F_{real}$ and $F_{comt}$, $F_{comp}$, $F_{com}$, and $F_{day}$ were 12.58 mmol m$^{-2}$ d$^{-1}$, 11.94 mmol m$^{-2}$ d$^{-1}$, 11.93 mmol m$^{-2}$ d$^{-1}$, and 46.32 mmol m$^{-2}$ d$^{-1}$, respectively. Thus, $F_{com}$ provided a more accurate estimation of $F_{real}$ than did $F_{day}$. The results indicate that the error associated with using diurnal data instead of diurnal–nocturnal data to calculate the $CO_2$ flux can be reduced by accounting for the nocturnal effect.

As the *SST* value was the most important factor influencing $pCO_{2w}$, the nocturnal effects of these parameters partially coincided. In contrast, no obvious diurnal–nocturnal relationship was observed for *Chl-a*; thus, the nocturnal effect of *Chl-a* was not considered in this study. Although the daily variation in the wind speed significantly affected the daily variation in the $CO_2$ flux, this parameter was not considered when we established the relationship of the nocturnal effect because 24 h wind data can usually be obtained.

The fitting effect of using the complete set of buoy data to build the $pCO_{2wd}$ model was poor; therefore, we chose to build the $pCO_{2wd}$ models based on data for the Pacific Ocean, Atlantic Ocean, and Indian Ocean, respectively. The Pacific and Atlantic Ocean sub-regions played major roles in the regional algorithmic model. The $pCO_{2wd}$ of the Indian Ocean was only related to $SST_d$, and the fitting results between $pCO_{2wd}$ and $SST_d$ were good. However, the algorithm for the Indian Ocean was only based on one station (BOBOA) from 2013 to 2017 because there was insufficient data for stations in the Indian Ocean. In the future, we hope to obtain more relevant data for the Indian Ocean to further improve the algorithmic modelling of this region.

The global $CO_2$ flux was calculated using the $pCO_{2wd}$ model and the established nocturnal effect. The source and sink areas of $CO_2$ in the global ocean were at low and high latitudes, respectively. The mean daily, monthly, and annual global $CO_2$ fluxes were $-4.80 \times 10^{-3}$ mmol m$^{-2}$ d$^{-1}$, $-23.36$ mmol m$^{-2}$ month$^{-1}$, and $-6.86$ mol m$^{-2}$ y$^{-1}$, respectively, indicating that the global ocean was an overall sink for atmospheric $CO_2$ from September 2020 to August 2021. During this period, the oceanic sources and sinks of $CO_2$ determined based on *cor_$F_{com}$* were smaller than those based on *day_$F_{com}$*. Compared with *day_$F_{com}$*, the global *cor_$F_{com}$* value was greater by 0.18 mmol m$^{-2}$ d$^{-1}$, thereby *day_$F_{com}$* overestimating the oceanic $CO_2$ sink by 10.21%. The mean monthly increase of *cor_$F_{com}$* was 2.50 mmol m$^{-2}$ month$^{-1}$, thus *day_$F_{com}$* overestimating the mean oceanic $CO_2$ sink by 10.68%. The mean annual increase of *cor_$F_{com}$* was 0.75 mol m$^{-2}$ y$^{-1}$, thus *day_$F_{com}$* overestimating the mean oceanic $CO_2$ sink by 10.89%.

In the current studies, the $pCO_{2W}$ algorithms were frequently built using data from small regions, and few algorithms were built from large areas. However, in order to estimate the global $CO_2$ flux using satellite data, a large-scale algorithm was used, which was not so accurate as the small-scale regional algorithms. We will improve the accuracy of

the global-scale $pCO_{2W}$ algorithm to further refine the process of estimating global daily $CO_2$ fluxes in future studies. The equation for calculating the $k$ used to determine the $CO_2$ flux is one of the many parameterised formulas that have been developed for establishing the relationship between the $k$ of $CO_2$ and wind speed. Different $k$ equations will yield different $CO_2$ fluxes. Although such differences were not considered in this study, we hope to address them in future studies.

**Author Contributions:** Conceptualization, T.Y. and R.J.; Methodology, T.Y. and R.J.; Software, R.J.; Validation, B.T., W.S. and S.H.; Formal Analysis, W.S.; Investigation, R.J.; Resources, T.Y.; Data Curation, R.J. and T.Y.; Writing—Original Draft Preparation, R.J. and T.Y.; Writing—Review & Editing, T.Y.; Visualization, R.J.; Supervision, B.T., W.S., S.H. and Y.W.; Project Administration, T.Y.; Funding Acquisition, T.Y. All authors have read and agreed to the published version of the manuscript.

**Funding:** This work was supported by National Natural Science Foundation of China (Grant No. 41906152, No. 42176012, No. 42130402), the Key Special Project for Introduced Talents Team of Southern Marine Science and Engineering Guangdong Laboratory (Guangzhou) (Grant No. GML2019ZD0602), the Global Change and Air-Sea Interaction II Program (Grant No. GASI-01-DLYG-WIND02 and No. GASI-01-DLYG-EPAC0) and the National Key Research and Development Program of China (2021YFC3101702).

**Data Availability Statement:** Publicly available datasets were analyzed in this study. These data can be found here: [https://www.ncei.noaa.gov/, accessed on 8 May 2022; https://cds.climate.copernicus.eu/, accessed on 8 May 2022; https://oceandata.sci.gsfc.nasa.gov/, accessed on 8 May 2022; https://disc.gsfc.nasa.gov/, accessed on 8 May 2022; https://resources.marine.copernicus.eu/, accessed on 8 May 2022].

**Acknowledgments:** We thank the Ocean Carbon Data System (OCADS) for providing the $pCO_2$, *SST*, and *SSS* data (https://www.ncei.noaa.gov/access/ocean-carbon-data-system/oceans/time_series_moorings.html), NASA for providing the *SST* and chlorophyll data and for making the data available systematically (https://oceandata.sci.gsfc.nasa.gov/directaccess/MODIS-Aqua/L3SMI/), the fifth-generation European Centre for Medium-Range Weather Forecasts (ECMWF) for providing the wind and atmospheric pressure data (https://cds.climate.copernicus.eu/cdsapp#!/dataset/reanalysis-era5-single-levels?tab=overview), Copernicus Marine Service for providing the *SSS* data (https://marine.copernicus.eu), and Earthdata for providing the atmospheric $CO_2$ and water vapour data (https://disc.gsfc.nasa.gov/datasets/SNDRAQIL3CDCCP_2/summary?keywords=CO2); accessed date: 8 May 2022.

**Conflicts of Interest:** The authors declare no conflict of interest.

## Appendix A

**Table A1.** Variable symbols in this article defined in the corresponding table.

| Quantity | Meaning |
| --- | --- |
| $pCO_{2w}$ | Partial pressure of $CO_2$ in seawater |
| $pCO_{2a}$ | Partial pressure of $CO_2$ in air |
| $F_{day}$ | The $CO_2$ flux calculated using diurnal buoy data |
| $F_{real}$ | The $CO_2$ flux calculated using diurnal–nocturnal buoy data |
| $F_{SST}$ | The daily $CO_2$ flux calculated using diurnal *SST* buoy data only, and other parameters except *SST* were diurnal–nocturnal buoy data |
| $F_{SSS}$ | The daily $CO_2$ flux calculated using diurnal *SSS* buoy data only, and other parameters except *SSS* were diurnal–nocturnal buoy data |
| $F_{k_{660}}$ | The daily $CO_2$ flux calculated using diurnal wind speed buoy data only, and other parameters except wind speed were diurnal–nocturnal buoy data |
| $F_{pCO_{2w}}$ | The daily $CO_2$ flux calculated using diurnal $pCO_{2w}$ buoy data only, and other parameters except $pCO_{2w}$ were diurnal–nocturnal buoy data |
| $F_{pCO_{2a}}$ | The daily $CO_2$ flux calculated using diurnal $pCO_{2a}$ buoy data only, and other parameters except $pCO_{2a}$ were diurnal–nocturnal buoy data |
| $pCO_{2wn}$ | The nocturnal $pCO_{2w}$. This variable was used to establish the nocturnal relationship using buoy data |

**Table A1.** *Cont.*

| Quantity | Meaning |
| --- | --- |
| $pCO_{2wd}$ | The diurnal $pCO_{2w}$. This variable was used to establish the nocturnal relationship using buoy data |
| $SST_n$ | The nocturnal $SST$. This variable was used to establish the nocturnal relationship using buoy data |
| $SST_d$ | The diurnal $SST$. This variable was used to establish the nocturnal relationship using buoy data |
| $F_{comp}$ | The $CO_2$ flux calculated using only the nocturnal effect for $pCO_{2w}$ and satellite data for each parameter |
| $F_{comt}$ | The $CO_2$ flux calculated using only the nocturnal effect for $SST$ and satellite data for each parameter |
| $F_{com}$ | The $CO_2$ flux calculated using only the nocturnal effect for $pCO_{2w}$ and $SST$ and satellite data for each parameter |
| $com\_pCO_{2wd}$ | $pCO_{2wd}$ calculated using remote sensing data of $SST_d$ and $Chl\text{-}a$ |
| $com\_pCO_{2wn}$ | The $pCO_{2wn}$ calculated using the nocturnal effect for $pCO_{2w}$ and $com\_pCO_{2wd}$ |
| $day\_F_{com}$ | The diurnal $CO_2$ flux calculated using diurnal remote sensing data of $SSS$, $pCO_{2a}$ and wind speed and remote sensing data of $SST_d$ and $com\_pCO_{2wd}$ |
| $cor\_F_{com}$ | The diurnal–nocturnal $CO_2$ flux calculated combining $com\_pCO_{2wd}$, $com\_pCO_{2wn}$, $SST_d$, and $SST_n$ with the diurnal–nocturnal remote sensing data of $SSS$, $pCO_{2a}$, and wind speed, considering the nocturnal effect |

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
