# Peer review of "Influence of the Nocturnal Effect on the Estimated Global CO2 Flux"

_remotesensing, doi:10.3390/rs14133192_

Round 1
Reviewer 1 Report
The topic of the article is topical in the light of the latest environmental programs of various directions. The role of the ocean in the absorption / release of various chemical compounds is huge. The work should be accepted for publication. There are some comments. Figure 4 shows the change in the concentration of CO2 in the water-air system. It is clear that this behavior depends on many parameters, which is well described by the authors in the introduction. Despite such large variations , the authors of the article write: The relationship established using a long time series of data could also be used to model missing data and improve the temporal resolution of the data (lines 409 – 410). A very controversial statement that contradicts the graph of Fig. 4 and the text in the introduction. It is necessary to either remove or evaluate. In conclusion, I suggest that the authors discuss the contribution to the "night effect" (in percentage terms) of the processes described in the introduction (references [7-16]).c
I looked through this article again. It discusses an important topic related to the absorption and release of carbon dioxide. New methods of calculating these processes with the division of the contribution of the night into the daily are proposed. This issue is very important from the point of view of remote optical methods, which are practically useless at night. It is clear that taking into account the night contribution is extremely important, since we do not know when the process of radiation or absorption begins. More biota. Her behavior is different at different times of the day. At night, many biological objects become active, which also leads to significant variations in these processes. The article is good, but it is necessary to continue working in this direction. Since other processes described well by the authors of the article in the introduction should be "drawn" into this absorption/radiation process. But in this form, the article can be published taking into account the comments of reviewers and editorial edits.
Reviewer 2 Report
Dear Editor,
Review of the paper: “Influence of the nocturnal effect on the estimated global CO2 flux”
In this paper a very interesting and important theme of investigation of temporal and spatial variabilities of CO2 flux in the Pacific, Atlantic and Indian Ocean sub-regions was analyzed.
The paper is based on a large amount of measured data taken from various projects worldwile and remote sensing technique. Methodology that combine “in situ” measurements and satellite remote sensing is promising.
The construction of the paper is good and correct. The suggestions bellow have the purpose to contribute with the authors.
My general comments and suggestions are as follows:
1. Page 2. Lines 85-88. The last paragraph on page 2 should be reformulated as a "work goal".
2. Page 3. Figure 1. Who is the author of Figure 1? If they are not the authors of the paper, please cite the source!
3. Page 4. Figure 3. The description of the figure is not clear enough.
4. Page 6. Figure 5. The description of the figure is not clear enough.
5. Page 6. Line 194. Flux Fk660 is not defined!
6. Page 9. Line 268. (a) nocturnal SST..
7. Page 9. Line 273. Is it (b)?
8. Page 9. Line 279. Is it (a)?
9. Page 11. Figure 12. The figure shows that the correlation is not linear. The correlation coefficient is small and should be commented on!
10. Page 12. Figure 13. The figure shows that the correlation is not linear. The correlation coefficient is small and should be commented on!
11. Page 15. Figure 17. Where is Figure 17 cited in the text? What is the meaning of this figure at this position of the text?
12. References should be checked. There are typographical errors and missing data.
Reviewer 3 Report
Understanding the diurnal-nocturnal variability of sea-air CO2 exchange is important for estimation global CO2 flux. This manuscript investigated the influence of the nocturnal effect by studying the relationship between the diurnal pCO2 and nocturnal pCO2, and a revised algorithm was proposed to improve the remote sensing of CO2 flux. This topic is quite unique. However, at this stage I think the whole structure of this study should be improved, and the accuracy of the inversion algorithm needs more validation.
I have some suggestions and comments and I hope these could be helpful to improve this study.
1. Some figures like Fig. 10, 16 and 17 might be not necessary. And Figure 17 wasn’t mentioned in the manuscript.
2. Some sentences are oddly written and there are too many signals for different parameters. A notation table might be helpful to discriminate the different abbreviations and signals. Prefix or suffix might be better than subscript.
3. Buoy data shown in Fig.4, Fig.7 and Fig.10 are from different stations. I guess that there should be many regional differences. Is the data at station Kodiak in Figure 4 is typical enough? Or just an example for confirming the diurnal-nocturnal differences? Information about Kiyomoto-Yoko experiment in 2003 was not shown in Fig.3. It’s not so precise to draw a conclusion that “no diurnal-nocturnal rule in Chl-a was observed (Line 284, Page 10)”. More data analysis is needed to support this point.
4. There are still large divergences for the relationship shown in Fig. 12 and 13, even though the regression could pass the 99.9% significant test. How can we determine the accuracy of this algorithm for estimating pCO2 and CO2 flux? More explanations are expected.
5. Page 9,Line 277. Nocturnal SST was obtained from diurnal SST using equation (6), and we can see that the differences are not so obvious. For remote sensing retrieval, can we use SST data observed by satellite at night?
Round 2
Reviewer 3 Report
I can see the improvement of this manuscript. Although the accuracy of algorithm is not the focus of this study, it will affect the result for quantitative analysis. I still recommend a statement for this point in the expectation part of the conclusion.
